# Regulation Games for Trustworthy Machine Learning

## Abstract

Existing work on trustworthy machine learning (ML) often focuses on a single aspect of trust in ML (*e.g.*, fairness, or privacy) and thus fails to obtain a holistic trust assessment. Furthermore, most techniques often fail to recognize that the parties who train models are not the same as the ones who assess their trustworthiness. We propose a framework that formulates trustworthy ML as a multi-objective multi-agent optimization problem to address these limitations. A holistic characterization of trust in ML naturally lends itself to a game theoretic formulation, which we call regulation games. We introduce and study a particular game instance, the SpecGame, which models the relationship between an ML model builder and regulators seeking to specify and enforce fairness and privacy regulations. Seeking socially optimal (*i.e.*, efficient for all agents) solutions to the game, we introduce ParetoPlay. This novel equilibrium search algorithm ensures that agents remain on the Pareto frontier of their objectives and avoids the inefficiencies of other equilibria. For instance, we show that for a gender classification application, the achieved privacy guarantee is $3.76\times$ worse than the ordained privacy requirement if regulators do not take the initiative to specify their desired guarantees first. We hope that our framework can provide policy guidance.

## 1 Introduction

The responsible deployment of machine learning (ML) models involves challenges beyond optimizing for model utility. New fields of study have emerged to address further reaching aspects of proper decision-making—collectively known as trustworthy ML. These include, among others, algorithmic fairness (Mehrabi et al., 2021), privacy (Liu et al., 2021), and robust ML (Szegedy et al., 2013). These objectives invariably present various trade-off with one another. Chang & Shokri; Suriyakumar et al. (2021) discuss the fairness-privacy trade-off, Kifer & Machanavajjhala considers utility-privacy. The fairness-utility trade-off is also widely studied (Wick et al.). Recently, works such as Esipova et al. (2022); Yaghini et al. (2023) considered the 3-way trade-off between fairness, privacy and utility.

Regardless of the method, prior work implicitly assumes that a *single* entity is in charge of implementing the different objectives. Unfortunately, regarding trust in ML purely from a single-agent lens runs the risk of producing trade-off recommendations that are *unrealizable* in practice. This is because nowadays ML models are trained, maintained, and audited by separate entities—each of which may pursue their own objectives. To carry out the aforementioned trade-off recommendations, it would require the agents to align their objectives and take coordinated action.

Given the multitude of agents and objectives involved, we argue that achieving trust in ML is inherently a multi-objective multi-agent problem (Rădulescu et al., 2020). Since Game Theory is the natural tool to model and analyze the interactions between different agents, we initiate the first study on multi-agent trustworthy ML within a novel class of games we call the *ML Regulation Games*. The concrete scenario that we study as a regulation game is the problem of specifying and enforcing fairness and privacy regulations for ML models. As an analogy, this is similar to emissions control agencies specifying and enforcing CO-2 emission levels on production vehicles In that market, regulators and car makers interact with each other and arrive at an acceptable and achievable cap on car emissions. In a similar fashion, in our proposed SpecGame for trustworthy ML, fairness and privacy regulators (who specify and enforce regulations) interact with the model builder (who trains

the ML model). The goal is to arrive at an acceptable (to regulators) and achievable (by the model builder) specification for the level of privacy and fairness guarantees.

We provide a formal characterization of the agents, their loss functions, strategies, and training outcomes. In doing so, we face our first unique challenge in specifying a game for trustworthy ML: characterizing the privacy regulator's loss function depends on a black-box estimation of the model's privacy guarantee, which has been theoretically shown impossibility (Gilbert & McMillan, 2018). We circumvent the issue of privacy loss estimation by pre-calculating a *Pareto frontier* (PF) between achieved fairness, privacy, and accuracy levels. This PF is then shared as *common-knowledge* (Fudenberg & Tirole, 1991) among all agents. Note that sharing knowledge does not require agents to take coordinated action (as is assumed in the single-agent setting), or otherwise preclude them from independent action in any way. Given the inherent optimality of the PF, calculation of the best-response functions (BRs), *i.e.* the minimizers of each agent's loss in response to others' actions, becomes independent of the privacy loss estimation. Eventually, the intersection of the BRs results in Nash equilibrium (NE) points. We call our resulting equilibrium-search algorithm ParetoPlay.

Note that, in general, not all equilibria are equally *societally beneficial*. Let us illustrate this using our previous analogy: imagine that the emissions regulatory has set a maximum tolerable amount of car emissions $x_0$. The car maker, being an independent agent, can choose not to follow that recommendation which allows it to save $y$ dollars in expected costs and release cars with emission levels $x > x_0$. The regulator in turn imposes a fine with dollar-amount value $C(x - x_0)$. Note that if $y > C(x - x_0)$, the car maker is incentivized to absorb the penalty as simply price of doing business. This results in an equilibrium state, but not a societally-beneficial one. We can potentially improve this situation by choosing an appropriate $C$; which is an example of *incentive design*. We formulate and empirically study incentive design for the SpecGame. Here, we take the role of the regulators who have analyzed a particular market (*e.g.*, facial recognition software) using ParetoPlay, and want to choose correct penalty scalers to enforce their desired guarantees. Our goal is to present a framework that would allow policy makers to push for more socially-beneficial equilibria, and thereby implement a more trustworthy ML in practice at a moderate cost to ML utility.

For our empirical evaluation, we instantiate SpecGame with four different tasks and two private and fair learning frameworks ((Yaghini et al., 2023; Esipova et al., 2022)) and set out to empirically answer three broad research questions: **(RQ1)** Can we characterize the suboptimality of reducing the inherently multi-agent trustworthy ML to the single-agent setting? **(RQ2)** What can the interactions between various agents tell us about the outcome of SpecGame? For example, how important is it for regulators to move first (or second) in the market? **(RQ3)** Can our study surface incentive design guidelines for policy makers?

In summary, we make the following contributions:

- We introduce a general framework for *ML regulation games* which enables us to characterize trust in ML as a multi-objective *and* multi-agent problem. We instantiate a concrete regulation scenario as SpecGame where fairness and privacy regulators attempt to introduce and enforce trustworthiness guarantees in a sensitive software market.

- We propose ParetoPlay, a novel equilibrium search technique that assumes shared knowledge of a pre-calculated PF between agent objectives. ParetoPlay allows us to efficiently simulate the interactions between agents in SpecGame, and recover equilibria points. Analysis of equilibria, in turn, enables us to tackle questions of *empirical incentive design*, as a mechanism to push for more societally-beneficial equilibria.

- Empirically, we highlight the suboptimality of studying trustworthy ML in a single-agent framework. For instance, we show that for a facial analysis software market, the achieved privacy guarantee is $3.76\times$ worse than the ordained privacy requirement if regulators do not take initiative to specify their desired guarantees to the market.

## 2 BACKGROUND

**Problem Setting: Specification.**   We consider a scenario involving an application of ML to automated facial analysis—similar to models considered in Krishnan et al. (2020). This is for illustrative purposes; our framework is generally applicable. Facial analysis models perform a range of sensitive

tasks, such as face recognition and prediction of sensitive attributes like gender or age.[1] It is clear that both fairness and privacy are relevant to the individuals that use such a model or contribute to its training data. In practice, regulators, such as governmental agencies or ethics committees, are put in charge of formulating and enforcing privacy and fairness requirements on the individuals' behalf.

**Game Theory.** A *Stackelberg* competition $\mathcal{G} = (\mathcal{A}, \mathcal{S}, \mathcal{L})$ is a game defined by a set of agents $\mathcal{A}$, their strategies $\mathcal{S}$, and their loss functions (aka, negative payoff functions) $\mathcal{L}$, where the players go in sequential order. In its simplest form with two agents, the *leader* initiates the game. Having observed the leader's strategy, a *follower* reacts to it. Analysis of Stackelberg competitions, involves solving a bi-level optimization problem where every agent is minimizing their loss subject to other agents doing the same. Solutions to the said problem recovers the *Nash equilibria* (NEs). Searching for NEs is intractable (PPAD-complete (Daskalakis et al., 2006)), which is why a super-set of them, known as *correlated equilibria*, have seen increasing attention due to ease with which they can be found (for instance, using polynomial weights algorithm) (Arora et al., 2012; Nisan et al., 2007).

In multi-objective optimization, and games in particular, we are interested in the *social welfare* (Shoham & Leyton-Brown, 2009) of an equilibrium to all agents. *Pareto efficiency* is a metric of social welfare, which adopted for the specification problem is (Yaghini et al., 2023):

**Definition 1** (Pareto Efficiency). *A model $\omega \in \mathcal{W}$, where $\mathcal{W}$ is the space of all models, is Pareto-efficient if there exists no $\omega' \in \mathcal{W}$ such that (**a**) $\forall i \in I$ we have $\ell_i(\omega') \leq \ell_i(\omega)$ where I is the set of objectives and $\ell_i \in \mathcal{L}$ is the loss of objective i; and that (**b**) for at least one objective $j \in I$ the inequality is strict $\ell_j(\omega') < \ell_j(\omega)$.*

**Privacy.** Differential Privacy (DP) (Dwork & Roth, 2013) is a mathematical framework that provides rigorous privacy guarantees to individuals contributing their data for data analysis. A typical DP mechanism is to add *controlled noise* to the analysis algorithm, making it difficult to identify individual contributions while still yielding useful statistical results. Formally, $(\varepsilon, \delta)$-differential privacy can be expressed as follows:

**Definition 2** (Approximate Differential Privacy). *Let $\mathcal{M} \colon \mathcal{D}^* \to \mathcal{R}$ be a randomized algorithm that satisfies $(\varepsilon, \delta)$-DP with $\varepsilon \in \mathbb{R}_+$ and $\delta \in [0, 1]$ if for all neighboring datasets $D \sim D'$, i.e., datasets that differ in only one data point, and for all possible subsets $R \subseteq \mathcal{R}$ of the result space it must hold that $\varepsilon \geq \log \frac{\mathbb{P}[\mathcal{M}(D) \in R] - \delta}{\mathbb{P}[\mathcal{M}(D') \in R]}$.*

DP, by definition, protects against worst-case failures of privacy (Dwork & Roth, 2013; Mironov, 2017). These failures are rare events and difficult to detect. As a result, sample-efficient estimation of DP guarantees is nearly impossible (Gilbert & McMillan, 2018). Prior work has shown though that it is still possible to estimate the privacy leakage of a model through an audit. For example, one can train multiple instances of the model to establish a lower-bound on the privacy privacy parameter $\varepsilon$ by training thousands of models (Tramer et al., 2022).

## 3 SPECGAME

We introduce SpecGame, an ML regulation game that captures the interactions between three agents involved in the life-cycle of an ML model (Tomsett et al., 2018): a *model builder* who is in charge of producing the model, and two regulators who are in charge of fairness and privacy of the resulting model, respectively. We note that our framework is general and can accommodate other objectives, as long as they are measurable with a cost function. We assume the model builder seeks to create the most accurate model. The regulators formulate the requirements and monitor the model for potential violations of their objectives based on recent regulations (Commission, 2021; Office of the Privacy Commissioner of Canada, 2019). The *fairness regulator* assesses the resulting model for potential discrimination using fairness metrics (Beutel et al., 2019), whereas the *privacy regulator* seeks to ensure that strong-enough guarantees exist to protect the privacy of the training data used

---

[1]The use of such models can, therefore, have severe ethical implications, motivating the need to optimize for their general trustworthiness, instead of focusing on one single aspect, such as their average utility.

for the model. We assume that regulators are able to give penalties[2] for violations of their respective objective which they formulate as a utility (or value) function.

Depending on whether regulators announce concrete specifications first, or if the model builder produces a model first with fairness and privacy guarantees of its choosing, we would have a game that is either *regulator-led*, or *builder-led*. This sequential order of interactions lends itself naturally to a Stackelberg competition (see Section 2). As agents interact with each other, the game is repeated: Figure 1 shows both a *regulator-led* (top) and a *builder-led* market (bottom). Since analysis of both settings is similar, without loss of generality (W.L.O.G), unless otherwise stated, we will assume a regulator-led market. In Section 5, we will discuss the impact of moving first. In this work, we do not consider a competition between regulatory bodies since both are assumed to be governmental agencies. We assume the regulators hold necessary information about the task at hand in the form of a Pareto Frontier (PF)[3] which they use to choose fairness and privacy requirements that taken together with the resulting accuracy loss are *Pareto efficient*: improving one objective would necessarily come at the cost of another (see Definition 1). This choice departs from the classical non-cooperative game formulations but we argue it is appropriate given that regulators do not seek to punish model builders for creating accurate models given that a well-generalized (robust) model is necessary for strong privacy (Li et al., 2021) and fairness guarantees (Kulynych et al., 2022).

## 3.1 AGENTS AND STRATEGIES

**Fairness Regulator.** The choice of a proper fairness measure is task-dependent. Hence, our regulation game framework does not make any assumptions on the applied metric. The fairness evaluation process takes as input a fairness metric $\Gamma_{\text{fair}}(\omega, D) : \mathcal{W} \times \mathcal{X} \mapsto \mathbb{R}^+$ (Barocas et al., 2018) chosen by the fairness regulator, the model $\omega \in \mathcal{W}$, and an adequate evaluation dataset $D_{\text{eval}} \in \mathcal{X}, D_{\text{eval}} \sim \mathcal{D}$, where $\mathcal{D}$ is the task's data distribution. The evaluation process then outputs $\widehat{\gamma}_\omega$ as an empirical estimate of the model's fairness violation.

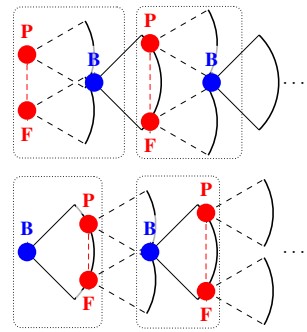

Assuming regulators lead, at the first stage of SpecGame, the fairness regulator's strategy is to specify the maximum tolerable fairness violation $\gamma \in \mathbb{R}^+$. Next, the builder creates the model $\omega$. At the end of the first stage, where losses are measured, the regulator's loss is

$$\ell_{\text{fair}}(\gamma; \omega) := \max\{0, \widehat{\gamma}_\omega - \gamma\}. \qquad (1)$$

Figure 1: **Repeated SpecGame between model Builder, and Privacy and Fairness regulators**—*(top)* regulators lead, *(bottom)* builder leads.

In the subsequent stages of the game, the regulator strategy is to announce a penalty (*e.g.*, a monetary fine) for violating its specification. We choose a linear form for such that the penalty is proportional to the excessive fairness violation ($\gamma_\omega - \gamma$). This formulation also follows the the common "expected utility hypothesis" in economics (Roberts, 1984). W.L.O.G. then, regulator's strategy is to choose $C_{\text{fair}} \in \mathbb{R}^+$ and announce the penalty $L_{\text{fair}}(\omega) = C_{\text{fair}} \ell_{\text{fair}}(\gamma; \omega)$. If $C_{\text{fair}}$ is too small, regulators strategy may not be effective in persuading the model builder to create a fairer model and thus lower $\widehat{\gamma}_\omega$. We discuss how to chose $C_{\text{fair}}$ in Section 4.2.

**Privacy Regulator.** We assume that the privacy regulator specifies its requirement within the framework of approximate DP (see Definition 2). To satisfy this requirement, the builder, has to adopt private training (Esipova et al., 2022; Yaghini et al., 2023) which bounds the privacy leakage of the model thus trained, and protects the privacy of sensitive training data, *e.g.*, such as individual data for facial analysis (see Section 2).

In multi-agent formulation of ML trustworthiness where separate entities train and audit the model, privacy estimation is an important part of privacy regulator's loss. However, in order to keep our framework general and future-proof, we abstract away the particular technique used for privacy

---

[2]As is customary in the economics literature, the penalties need not be monetary. It is sufficient that they present a viable risk. This, for instance, may take the form of expected lost revenue due to a watch-dog (or NGO) reporting on fairness and/or privacy violations of an ML-as-a-Service platform.

[3]It has been shown that this does not require access to private data since access to data from the same domain is sufficient to calculate the PF Yaghini et al. (2023).

estimation. We note that the technique used bears importance in the simulation of the game, but we defer that discussion to Section 4 where we discuss our simulator for SpecGame. Concretely, we treat privacy auditing mechanism as a function that, at a fixed $\delta$, outputs an estimate of $\widehat{\varepsilon}_\omega \in \mathbb{R}^+$, *i.e.*, privacy of the model implemented by model builder. Based on this estimate and the privacy parameter required from the privacy regulator $\varepsilon \in \mathbb{R}^+$, the privacy-related loss term can be formulated as

$$\ell_{priv}(\omega) = \begin{cases} \exp\left(\widehat{\varepsilon}_\omega - \varepsilon\right) & \widehat{\varepsilon}_\omega > \varepsilon \\ 0 & \text{otherwise} \end{cases} \tag{2}$$

The privacy loss in Equation (2) is formulated as a exponential because the privacy parameter $\varepsilon$ is similarly defined in Definition 2. As a result, the privacy loss can be upper-bounded by probability ratios of indistinguishability (see Section C.1) Note that the privacy loss is positive only when $\varepsilon < \widehat{\varepsilon}_\omega$ and zero if the model builder followed or over-accomplished the privacy regulator's guidelines. Similarly to the fairness regulator, the privacy regulator scales its loss with $C_{priv} \in \mathbb{R}^+$, announcing $L_{priv}(\omega) = C_{priv}\ell_{priv}(\omega)$ as penalty to the model builder.

**Model Builder.** The model builder is in charge of implementing an ML task, such as facial recognition, by creating a model $\omega \in \mathcal{W}$ where $\mathcal{W}$ is the space of all possible models. While their primary focus is achieving high model utility (*e.g.*, accuracy), they are aware that the final trained model will have to face regulatory audits—either before deployment during internal audits, or in production with an external regulator. In particular, the builder knows that fairness or privacy violations can lead to penalties from the regulators. Therefore, the respective risks have to be included into the model builder's utility function, either as constraints, or as additive penalties to the loss:

$$L_{build}(\omega) = \ell_{build}(\omega; D_{\text{test}}) + \lambda_{priv}L_{priv}(\omega) + \lambda_{fair}L_{fair}(\omega) \tag{3}$$

where $\lambda_{priv}, \lambda_{fair} \in \mathbb{R}^+$. That is, the model builder's overall loss is its model loss $\ell_{build}(\omega)$ on a test dataset $D_{\text{test}} \sim \mathcal{D}$ plus the penalties incurred from privacy and fairness regulators ($L_{priv}$ and $L_{fair}$).

## 3.2 GAME FORMALIZATION, BEST RESPONSE AND NASH EQUILIBRIA

Formally the SpecGame $\mathcal{G}$ is a repeated Stackelberg game with the interaction stage defined as $\mathcal{G}_{\text{stage}} = (\mathcal{A}, \mathcal{S}, \mathcal{L})$ (shown in Figure 1 as dotted interaction windows), with the set of agents $\mathcal{A} = \{build, fair, priv\}$, losses $\mathcal{L} = \{\mathcal{L}_{build}, \mathcal{L}_{fair}, \mathcal{L}_{priv}\}$, and the strategy space $\mathcal{S} = \mathcal{S}_{build} \times \mathcal{S}_{fair} \times \mathcal{S}_{priv}$ as defined earlier. The overall *discounted* loss of agent $i \in \{build, fair, priv\}$ is $L_i = \sum_{t=1}^{\infty} \left\{ \prod_{i=1}^{t} c_i^{(t)} \right\} L_i^{(t)}$ for agent loss $L_i \in \mathcal{L}_i$ where $t$ is the time index. From a game theory perspective the discounting factor $c_i^{(t)} \in [0, 1]$ represent the fact that agents care about their loss in the near-term more than in the long run (Shoham & Leyton-Brown, 2009). W.L.O.G, we assume a constant discounting factor for each agent $c_i^{(t)} := c \in [0, 1] \,\forall t$, therefore, $L_i = \sum_{t=1}^{\infty} c^t L_i^{(t)}$. To analyze the SpecGame, we would typically need to calculate the best-response (BR) maps of agent strategies. For instance, the model builder's BR is the strategy $s_{build}^* \in \mathcal{S}_{build}$ that minimizes $L_{build}$ assuming regulators are also choosing their best responses. If every agent is choosing their best response strategy, we recover an NE (see Appendix F for details).

## 4 PARETOPLAY: BEST-RESPONSE PLAY ON THE PARETO FRONTIER

The SpecGame described in Section 3 cannot be easily simulated directly due to challenges in forming the agents' loss functions, most notably, because the privacy cost of a trained model is difficult to estimate in a black-box way post training (see Section 1). However, even if future developments make privacy estimation more efficient, the highly non-convex nature of agent losses in SpecGame makes them intractable for a typical game theoretic analysis which involves calculating and reasoning about its NEs. More importantly, although NEs are optimal w.r.t. single-agent deviations, they are often not Pareto efficient (see Definition 1). For example, seeking NEs can provide 'solutions' where both the model builder and a regulator's losses can be improved simultaneously.

We introduce ParetoPlay to address these problems by assuming agents share as common-knowledge a pre-calculated PF between privacy, fairness, and model utility. This assumption, in turn, has the advantage of restricting the search for equilibria to correlated equilibria that are likely to be on the PF. ParetoPlay is a general algorithm that can be run to simulate a SpecGame for any ML algorithm as long as a PF can be obtained.

**Using the PF as a correlation device.** Prior work has shown that fairness, accuracy, and privacy objectives are correlated. Well-generalized (*i.e.*, accurate) models have been shown to have better privacy and fairness guarantees (Bagdasaryan et al., 2019; Li et al., 2021), and that overfitting and memorization is a root cause of privacy leakage (Shokri et al., 2017) and disparate fairness impact (Kulynych et al., 2022). Additionally, it has been empirically shown (Yaghini et al., 2023) that *the PF is task-dependent as opposed to being data-dependent.* This means that every agent can estimate the PF on its own, without access to other agents' data. Thus, assuming a common-knowledge of the PF between agents is realistic. We also empirically verify this assumption in Appendix J with agents calculating the PF and playing ParetoPlay on separate datasets. Sharing the PF has important implications for the equilibrium search: the Pareto frontier gives a signal to every player what to play (similar to how a stop-light allows drivers to coordinate when to pass an intersection). This is known as a *correlation device*. If playing according to the signal is a best response for every player, we can recover a correlated equilibrium (CE).

**Making a uniform strategy space.** In the regulator-led SpecGame repeated for $n$ rounds. Note that SpecGame is an infinitely-repeated game, therefore, $n \to \infty$. We will provide convergence results in Section 4.1. The strategy set for either regulator is $\mathcal{S}_{reg} = \mathcal{S}_{reg}^{(1)} \times \prod_{t=2}^{n} \mathcal{S}_{reg}^{(t)}$ for $reg \in \{fair, priv\}$. In the first round, the strategy set $\mathcal{S}_{reg}^{(1)}$ is to announce a specification for their parameter (*i.e.*, $\gamma, \varepsilon$). However, in the proceeding $n-1$ rounds, $\mathcal{S}_{reg}^{t} = \mathcal{S} \quad \forall t > 1$ the strategy is to announce fines $\mathcal{S} = \{L_{reg} \in \mathbb{R}^+\}$. To simulate and reason about the SpecGame, it is beneficial to make the regulators' strategy set consistent across all stages. That is, we want to make the domain of strategies in every stage to be the same. This allows us to have a single stage game that is repeated. To do so, W.L.O.G., for $t > 1$, we assume a mapping from the penalty values in $\mathcal{S}$ to trustworthy parameters values $s_{reg} = (\gamma, \varepsilon)$ used by the model builder in the preceding round that caused the penalty. Furthermore, in ParetoPlay, where we additionally assume access to PF, we can make a similar assumption for the model builder: there exists a mapping from models to trustworthy parameters on the PF. Therefore, whenever the model builder announces a model, we can assume they have announced their chosen trustworthy parameters $s_{build}$. This allows us to write the strategy space of all agents and across all stages as $\mathcal{S} = \mathcal{S}_{stage}^n$ where $\mathcal{S}_{stage} = \mathcal{S}_{fair} \times \mathcal{S}_{priv} \times \mathcal{S}_{build} = \{(s_{fair}, s_{priv}, s_{build}) \mid s_i = (\gamma, \varepsilon), \gamma \in [0, 1], \varepsilon \in \mathbb{R}^+, i \in \{fair, priv, build\}\}$.

## 4.1 SIMULATING SPECGAME WITH PARETOPLAY

The game starts by distributing a PF between all agents. The PF is formed by training multiple instances of the chosen ML models in $R = \{\omega(s) \mid \omega \in \mathcal{W}\}$ before the game using different guarantee levels $s := (\gamma, \varepsilon)$ and then calculating the PF $P = \mathrm{PF}(R)$ where $\mathrm{PF} : [0, 1] \times \mathbb{R}^+ \mapsto [0, 1] \times \mathbb{R}^+ \times \mathbb{R}^+$ is a map from guarantee levels to a tuple of fairness, privacy and builder losses.

The agent who takes the first step, chooses the initial specification of the parameters $s$. Assuming regulators go first, given the no-compete assumption between them, they may choose to

---

**Algorithm 1 ParetoPlay**: Regulator-led

**Input:** Initial PF input $R^{(0)}$, total number of game rounds $T$, agents $\{A_{build}, A_{reg} \mid reg \in \{fair, priv\}\}$, step size $\eta$
1: **for** $t \in \{0, 1, \ldots, T\}$ **do**
2:      $P \leftarrow \mathrm{PF}(R^{(t)})$           ▷ Estimate the PF
3:      **if** t = 0 **then**          ▷ First round of the game
4:          $s^0 \leftarrow \textsc{ChooseSpec}(P, \{A_{fair}, A_{priv}\})$
5:      **else if** $t \mod 3 \neq 1$ **then**      ▷ Regulators moves
6:          $C_{reg}^{(t)} \leftarrow \textsc{ChoosePenaltyScale}(s^{(t-1)})$
7:          $s^{(t+1)} \leftarrow s^{(t)} - \eta \left\langle e_{reg}, \nabla_s L_{reg}^P(s^{(t)}; C_{reg}^{(t)}) \right\rangle$    ▷ reg only adjusts its own parameter
8:      **else**              ▷ Builder move
9:          $s^{(t+1)} \leftarrow s^{(t)} - \eta \nabla_s \ell_{build}^P(s^{(t)})$
10:      $R^{(t+1)} \leftarrow R^{(t)} \cup \{\textsc{Calibrate}(\omega, s^{(t+1)})\}$
11:      $\eta \leftarrow c_i \cdot \eta$      ▷ $A_i$ discounts its payoff by $c_i$ (decay factor)
12: **Output** $s^{(T)}$

---

jointly select a point on the Pareto Frontier and use it for the initial constraints. If they do so, they effectively become a combined regulator with a loss $\ell_r = \lambda_r \ell_{fair}(s) + (1 - \lambda_r)\ell_{priv}(s)$, for some $\lambda_r \in [0, 1]$ which decides the trade-off between fairness and privacy that the combined regulator seeks (ChooseSpec(.) in Algorithm 1). We note however, that from the second stage onward, each regulator interacts independently with the model builder. If the model builder moves first, it would select a point that would minimize its own loss function $\ell_{build}(s)$. In Section 3.2, we mentioned that in a repeated game, agent $i$ discounts their past losses over time by $c_i$. From an optimization point of view, $c_i$ appears as a decay factor (Line 11 in Algorithm 1).

**Choosing penalty scalers.**   Consider builder's step at $t + 1$ as seen by the regulators,

$$\boldsymbol{s}^{(t+1)} = \boldsymbol{s}^{(t)} - \eta \nabla_{\boldsymbol{s}} \ell_{build}(\boldsymbol{s}^{(t)}) - \eta C_{priv} \nabla_{\boldsymbol{s}} \lambda_{priv} \ell_{priv}(\boldsymbol{s}^{(t)}) - \eta C_{fair} \nabla_{\boldsymbol{s}} \lambda_{fair} \ell_{fair}(\boldsymbol{s}^{(t)}), \qquad (4)$$

where $\nabla_{\boldsymbol{s}}$ denotes the gradient with respect to the trustworthy parameters $\boldsymbol{s}$. Note the penalty scalars. $\lambda_{reg}$ are private information to the builder: a company is not incentivized to disclose how much a governmental penalty would be affecting its decisions. But as Algorithm 1 shows, regulators do not need to know $\lambda_{reg}$ to impact the builder's loss and force it to change strategy. This is because the penalty scalars $C_{reg}$ chosen by the regulators can create the same effect. In the next section, we use these scalars to incentivize more societally desirable equilibria.

**Approximation and calibration.**   In ParetoPlay, we estimate all agent losses on the PF $P$ (hence the $\ell_*^P$ in Algorithm 1). Our estimation involves a linear interpolation on $P$. Interpolation may lead to estimation errors, as the estimated next parameters $\boldsymbol{s}^{(t+1)}$ may, in fact, not be on the PF. Given that the PF is shared, this can lead to compounding errors, and non-convergent behavior. We avoid this by including a *calibration* step at the end of each round. In CALIBRATE in Algorithm 1, we train a new model using the chosen parameter $\boldsymbol{s}^{(t+1)}$, and add new objective loss results to the prior result set $R$. A new, potentially improved PF is recalculated next round with the new results (line 2). Next we show that ParetoPlay induces a correlated equilibrium:

**Theorem 1.** *ParetoPlay recovers a Correlated Nash Equilibrium of the SpecGame.*

We differ the proof to Section K. We also provide a proof sketch for the convergence of ParetoPlay in Section K.1, where we see ParetoPlay as essentially a sub-gradient optimization procedure with square-summable but not summable step sequences.

## 4.2   INCENTIVE DESIGN: HOW TO SET PENALTIES $C_{fair}$ AND $C_{priv}$

ParetoPlay ensures that the found equilibria found are on the Pareto frontier and, therefore, are efficient. However, as we established, not all such equilibria are desirable to the regulators. The "cost-of-doing" scenario we noted in Section 1 where regulators' penalties are not enough to effect a change in model builder's behavior is an example of such an equilibrium. If the regulators seek to escape such an equilibrium, they can do so by adjusting the penalty scalars $C_{fair}$ and $C_{priv}$ (hereon, $C_*$). If the regulators had no consideration for the model builder's loss, then their best strategy should be to choose a very large $C_*$; but this runs the risk of disincentivizing participation completely by increases the risk of insurmountable penalties for the builder. So how should we choose $C_*$?

**Reduction to a multi-objective problem.**   Given the incomplete information of the regulator about the loss of the model builder ($\lambda_{fair}$ and $\lambda_{priv}$ are unknown to the corresponding regulator), an exact answer is not possible. However, we can find good candidates for $C_*$ by reducing the multi-agent problem, to that of a multi-objective problem. This means that the regulators need to consider model loss as one of *their own* objectives and recover the Pareto frontier (PF) between all three objectives.

There exist standard techniques to recover the Pareto frontier of a multi-objective optimization problem (which always exists for any feasible problem). *Scalarization* (Boyd & Vandenberghe, Section 4.7.4) is such a technique that provided each objective is convex, can recover all of the Pareto frontier, and if not; at least a part of it. In Appendix C.2, we provide details on how we can leverage scalarization to obtain good $C_*$.

## 5   EXPERIMENTAL RESULTS

We instantiate ParetoPlay with two algorithms: FairPATE (Yaghini et al., 2023) and DPSGD-Global-Adapt (Esipova et al., 2022). Both algorithms train fair and private classification models and adopt approximate DP as their privacy notion. For each, we choose an appropriate strategy vector space $\boldsymbol{s}$ that the agents update throughout the game (see Section 4 on making the strategy space uniform). Different values for $\boldsymbol{s}$ train models with different levels of fairness and privacy guarantees. FairPATE measures fairness through maximum demographic parity gap; so we define $\boldsymbol{s}_{\text{FairPATE}} := (\gamma, \varepsilon)$, *i.e.*, the privacy budget and the maximum tolerable demographic fairness gap. DPSGD-Global-Adapt, on the other hand, measures fairness through maximum accuracy gap between subgroups as a result of using DP (*i.e.*, disparate impact of DP). We define $\boldsymbol{s}_{\text{DPSGD-Global-Adapt}} := (\tau, \varepsilon)$ which constitutes

fairness tolerance threshold $\tau$ and privacy budget $\varepsilon$. Before every game, we train multiple instances of the models by varying $s$ to calculate the PF. We defer the details to Section G.

**Experimental Setup.** We simulate the games on UTKFace, FairFace, CelebA, and MNIST datasets for 20 rounds. All games are regulator-led unless specified otherwise. After each round of the game, we log the current strategies $s$, the achieved privacy cost $\varepsilon$, the achieved maximum fairness gap (Disparity), as well as accuracy. FairPATE produces coverage as another measure of utility that denotes the percentage of queries answered by the model at inference time. Our research questions (RQs) are as follows: (RQ1) if regulators produced specifications for trustworthy parameters $s$ using a single-agent-optimized model, would that recommendation be respected in the multi-agent setting? (RQ2) How do the agents interact with each other during the game in different setups? (RQ3) What strategies can regulators use to achieve a desired equilibrium?

**RQ1: Single-agent-optimized specifications lead to sub-optimalities in the multi-agent settings**
. Game dynamics can converge to equilibria that violate regulators' desired guarantees. We show examples of the resulting trust gap in Figure 2 with both FairPATE and DPSGD-Global-Adapt. In all four games, the regulators act first and choose strategies $s$ that produce models with desired $\varepsilon(s)$ and maximum fairness gap $\Gamma(s)$. However, we can observe in Figure 2 that each game convergences to a point that either violates the fairness constraint or the privacy constraint, *i.e.*, final models all have trust gaps. To address this, regulators should adjust their incentives which we discuss in RQ3.

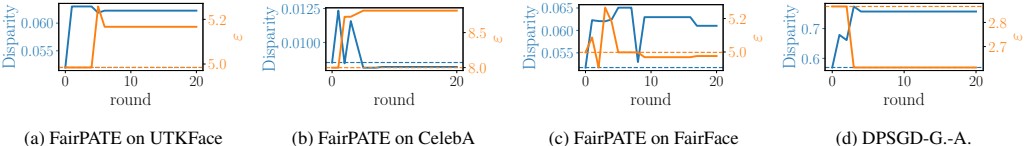

| (a) FairPATE on UTKFace | (b) FairPATE on CelebA | (c) FairPATE on FairFace | (d) DPSGD-G.-A. |

Figure 2: **ParetoPlay dynamics for regulator-led SpecGame.** We show fairness and privacy objective values in blue and orange, respectively. The constraints on each objective is set at round=0 and is constant throughout the simulation (marked by a dashed line). In (a) and (b), the privacy constraints are violated. In (a), (c), and (d), the fairness constraints are violated. Given that the initial specification is always on the PF, these show that applying single-agent recommendations to multi-agent setups leads to sub-optimal equilibria.

**RQ2: SpecGame leader has a first-mover advantage.** In traditional Stackelberg competitions, the first-mover has an advantage (Fudenberg & Tirole, 1991). We demonstrate that this also holds in the case of SpecGame. Recall that in each game, the first-mover chooses the point on the Pareto surface that minimizes their loss. All other parameters in both games, including regulators' fairness and privacy constraints, remain the same throughout the game run. Consider Figure 3. When the builder leads, they choose a starting point with large $\varepsilon(s)$ and $\Gamma(s)$ to maximize accuracy and coverage. The game converges to a point with high accuracy and coverage as well; favouring the model builder. Similarly, with smaller $\varepsilon(s)$ and $\Gamma(s)$, the equilibrium of the regulator-led is more desirable to the leader.

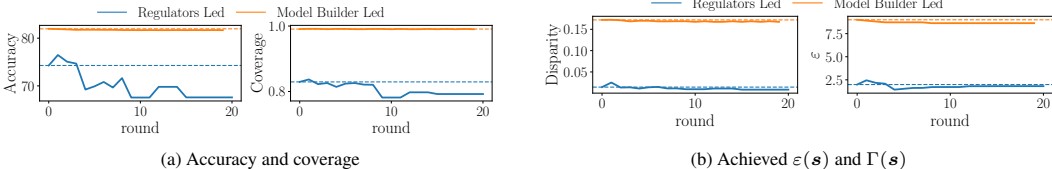

| (a) Accuracy and coverage | (b) Achieved $\varepsilon(s)$ and $\Gamma(s)$ |

Figure 3: **First-mover has an advantage in SpecGame.** We compare regulator-led and builder-led games on UTKFace with all other hyperparameters kept constant. The regulator-led game achieves lower privacy cost and fairness violation, while builder-led game achieves higher accuracy and coverage—demonstrating an equilibrium that favors the first mover.

**RQ3: Regulators can enforce desired equilibria.** Ideally, a game should converge to a point with strategies $s^T$ very close to what the regulators choose to satisfy regulators' constraints while preserving model utility. First, we demonstrate that the convergence points are influenced by the scale of regulator penalties. In Figure 4a, we show games with the same starting point but different regulator penalty scalars $C_{fair}$ and $C_{priv}$. Games with higher penalties tend converge to points with lower model utilities. This shows that games with high regulator penalties favor the regulators more.

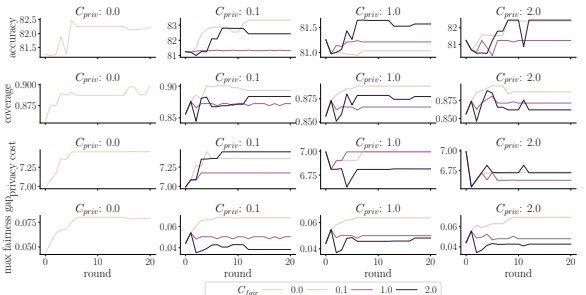 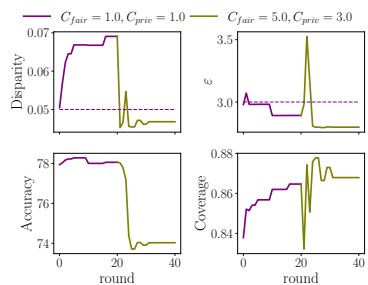

(a) **Regulators' losses need to be weighed at least similarly to the builder's for trustworthiness constraints to be enforceable.** Experiments on UTKFace assuming $\lambda_{fair} = \lambda_{priv} = 1$, with the initial constraints of $\gamma_0 = 0.05$ and $\epsilon_0 = 3.0$ kept constant across the simulation as before. Colors in the figure correspond to different $C_{fair}$ values, while columns reflect $C_{priv}$ values. To ensure the fairness and privacy constraints are satisfied, $C_{fair}$ and $C_{priv}$ need to be around 1; that is, comparable to builder's loss.

(b) **Regulators can enforce desired equilibria.** Each simulation consists of two stages. In the first stage, regulators set initial constraints (marked with dashed lines) which builders exceed. At this point, in the second stage, the regulators adjust the penalty scalars $C_{fair}$ and $C_{priv}$ which allows them to successfully enforce the initial constraints. Albeit at a cost to builder's accuracy but interestingly not coverage.

Regulators can change their incentives after the game has converged to find a more desirable equilibrium. We demonstrate this in Figure 4b in which the game has two stages. In the first stage, as before, regulators set initial constraints (namely, $\varepsilon \leq 3$ and $\gamma \leq 0.05$). The fairness constraint is not satisfied by the model builder until round 20 and the game has reached an equilibrium. At this point, in the second stage, regulators adopt higher penalties to address the fairness violation (from $C_{fair} = 1$ to $C_{fair} = 5$) which forces the builder to respect the constraint.

## 6 RELATED WORK

It has been shown that private training of ML models negatively influences utility (Tramer & Boneh, 2020) and fairness (Suriyakumar et al., 2021; Farrand et al., 2020). A general line of work on integrating those objectives by adapting the training procedure (Xu et al., 2019; Mozannar et al., 2020; Franco et al., 2021; Tran et al., 2021), or identifying favorable trade-offs between subsets of these objectives (Avent et al., 2019) has emerged over the past years. Note that, in contrast to our work, all prior frameworks to unify different objectives or characterize trade-offs do not consider the inherent multi-agent nature of the problem. Yet, we can leverage their methods to instantiate our regulation games. In this work we build on to recent frameworks, namely FairPATE (Yaghini et al., 2023) and DPSGD-Global(-Adapt) (Esipova et al., 2022). FairPATE extends the private aggregation of teacher ensemble (PATE) algorithm of Papernot et al. (2018) with an unfairness mitigation. The adaptive clipping framework modifies the differential private stochastic gradient descent algorithm (DP-SGD) of Abadi et al. (2016) with disparate clipping for data points from different sub-groups. The fact that our regulation games can be instantiated with two frameworks that differ so significantly in their approach to integrate privacy and fairness highlights the universality of our work. In work closest to ours, Jagielski et al. (2019) study the trade-offs between privacy, fairness, and accuracy within a game theoretic framework through a two-player zero-sum game. Our focus is on formulating the regulation game, with the purpose of designing proper incentive. We are interested in the more general (and more realistic) case of having multiple agents (such as two regulatory bodies) interacting with the model owner which does not admit a two-player zero sum game solution.

## 7 CONCLUSION AND FUTURE OUTLOOK

We introduced a general framework to study trustworthy ML models in multi-objective multi-agent scenarios through Regulation Games. We provided a concrete instantiation, SpecGame, and an equilibrium search algorithm ParetoPlay which allowed us to simulate the outcomes of policy announcement for minimum fairness and privacy requirements and to to discuss empirical incentive design and provide policy guidelines. Our regulation games can be extended to include more actors (*e.g.*, other regulators) and consider information asymmetries between model builder and the regulators (see Section J) and thus be applicable in an even broader range of practical scenarios.

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

## A  NOTATION

| Notation | Explanation |
|:---:|:---|
| $S_i$ | Strategy (decision) set of actor $i$ |
| $\omega$ | Model |
| $\Gamma$ | Fairness metric |
| $\ell_{fair}(\gamma; \omega)$ | Fairness regulator loss of model $\omega$ with constraint $\gamma$ |
| $\gamma$ | Fairness gap |
| $\widehat{\gamma}(\omega)$ | Measured fairness gap on model $\omega$ |
| $\ell_{priv}(\omega)$ | Privacy regulator loss of model $\omega$ |
| $\varepsilon$ | Privacy cost |
| $\widehat{\varepsilon}(\omega)$ | Estimated privacy cost on model $\omega$ |
| $\ell_{build}(\omega; D)$ | Model builder (utility) loss on dataset $D$ |
| $s$ | strategy |
| $C_*$ | regulator penalty scalar |
| $\eta$ | step size used by model builder for updates |
| $c_*$ | step size discounting factor |

Table 1: Table of Notation

## B  CHARACTERIZING EQUILIBRIA QUALITY WITH OPTIMALITY GAPS

In the single-agent optimization, the regulators, and model builder work as one centralized entity. The central entity can choose a point $s_{\text{OPT}}$ on the PF. By definition, using $s_{\text{OPT}}$ we cannot improve any objective without hurting another. In the multi-agent setup, however, even assuming we start from $s_{\text{OPT}}$, we will likely not stay at $s_{\text{OPT}}$ since, despite sharing the PF as a correlation device, every agent makes independent decisions. This causes suboptimalties w.r.t. $s_{\text{OPT}}$ that we wish to characterize.

An achieved equilibrium may not be desirable to one or both regulators. This may be due the model builder choosing to accept unfair, or non-private models and simply absorb the penalties as price of doing business; and therefore rendering the penalty mechanism ineffective. In this case, auditors can choose to increase their scalers $C_{fair}$ and $C_{priv}$ accordingly. Another reason may be that the values $\gamma$ and $\varepsilon$ were mis-specified. In other words, the prior values may be too harsh or unrealistic for the given task and data. Given that in ParetoPlay we empirically train and evaluate the model to form the PF, we avoid the latter case. Therefore, we focus on the former case, and provide a bespoke characterization for ParetoPlay where penalties may be ineffective, or unnecessarily harsh.

We note that there are other standard characterizations for social welfare (such as Price of Anarchy (Shoham & Leyton-Brown, 2009)). However, they are often evaluated theoretically, which is a challenge for empirical simulation of ParetoPlay. We leave the theoretical characterization of SpecGame to future work, and focus on the following characterization which allows us to empirically study questions of incentive design in Section 5.

In the start of the regulator-led ParetoPlay game, regulators choose the initial strategy to be $s^0$ according to their constraints, producing loss profile $(\ell_{fair}(s^0), \ell_{priv}(s^0), \ell_{build}(s^0))$. The game then converges to $s^T$ with $(\ell_{fair}(s^T), \ell_{priv}(s^T), \ell_{build}(s^T))$. If $s^0$ and $s^T$ are the same, the regulators' constraints are satisfied. If not then one of the following scenarios must be true: **(a)** $\ell_{fair}(s^T) \leq \ell_{fair}(s^0)$ and $\ell_{priv}(s^T) \leq \ell_{priv}(s^0)$: Both privacy and fairness constraints are satisfied, but $\ell_{build}(s^T) > \ell_{build}(s^0)$ must be true as well since both points are on the PF. The loss in model utility (*i.e.*, accuracy) constitutes a utility optimality gap, or *utility gap* for short. The result may still be acceptable if the utility gap is small since the model builder can seek better model designs to remedy the gap. **(b)** Either $\ell_{fair}(s^T) > \ell_{fair}(s^0)$ or $\ell_{priv}(s^T) > \ell_{priv}(s^0)$: One of the constraints are violated. This constitutes a trustworthiness optimality gap, or *trust gap*, which is not acceptable to the regulators. In this case, regulators may seek to adjust their penalty scalers to ensure their requirements are met. This the topic of incentive design which we consider next.

## C    Miscellanous details

### C.1    Modeling the Privacy Regulator's

Consider the approximate DP definition:

**Definition 3** (Approximate Differential Privacy). *Let $\mathcal{M} \colon \mathcal{D}^* \to \mathcal{R}$ be a randomized algorithm that satisfies $(\varepsilon, \delta)$-DP with $\varepsilon \in \mathbb{R}_+$ and $\delta \in [0, 1]$ if for all neighboring datasets $D \sim D'$, i.e., datasets that differ in only one data point, and for all possible subsets $R \subseteq \mathcal{R}$ of the result space it must hold that $\varepsilon \geq \log \frac{\mathbb{P}[\mathcal{M}(D) \in R] - \delta}{\mathbb{P}[\mathcal{M}(D') \in R]}$.*

Assume $\delta = 0$. If $\widehat{\varepsilon}_\omega = \frac{P[\mathcal{M}(\tilde{D}) \in R]}{P[\mathcal{M}(\tilde{D}') \in R]}$ for some datasets $\tilde{D}, \tilde{D}'$ where the largest difference between empirical probability measures $P(.)$ is calculated.

$$\exp\left(\widehat{\varepsilon}_\omega - \varepsilon\right) = \frac{\exp(\widehat{\varepsilon}_\omega)}{\exp(\varepsilon)} \tag{5}$$

$$\geq \frac{P\left[\mathcal{M}(\tilde{D}) \in R\right]}{P\left[\mathcal{M}(\tilde{D}') \in R\right]} \cdot \frac{\mathbb{P}\left[\mathcal{M}(D') \in R\right]}{\mathbb{P}\left[\mathcal{M}(D) \in R\right]} \tag{6}$$

$$= \frac{P\left[\mathcal{M}(\tilde{D}) \in R\right]}{\mathbb{P}\left[\mathcal{M}(D) \in R\right]} \cdot \frac{\mathbb{P}\left[\mathcal{M}(D') \in R\right]}{P\left[\mathcal{M}(\tilde{D}') \in R\right]} \tag{7}$$

If the probability estimation of $P(.)$ over $\tilde{D}$ are a good model for $\mathbb{P}$ and $D$, i.e., $P[\mathcal{M}(\tilde{D}) \in R] = \mathbb{P}[\mathcal{M}(D) \in R]$) then the right hand side of eq. (7) is 1; but this means that $\widehat{\varepsilon}_\omega = \varepsilon$ and the privacy loss of the regulator will be 0. If the empirical probabilities are underestimating the true probabilities, then the regulator loss will also be positive and will scale with ratio with which the probabilities are underestimated.

### C.2    Scalarization for Setting Penalties

We leverage scalarization to find good $C_*$. For our problem, the objective loss of scalarized problem is $\min_{\boldsymbol{s}} \alpha_1 \ell_{build}(\boldsymbol{s}) + \alpha_2 \ell_{fair}(\boldsymbol{s}) + \alpha_3 \ell_{priv}(\boldsymbol{s})$    $(\star)$ where $\boldsymbol{s}$ is the set of hyper-parameters, e.g., $\boldsymbol{s} = (\gamma, \varepsilon)$. We implicitly assume $\ell_{build}(\boldsymbol{s})$ is always optimized w.r.t. model weights given a particualr $\boldsymbol{s}$. $\alpha_i \geq 0, i \in [3]$ is a free parameter. Different choices for $\alpha_i$'s will give us various points on the Pareto frontier. Under the assumption of convexity then, all the steps in ParetoPlay are minimizers of the scalarized problem. Matching $(\star)$ with Equation (4) (with privacy and fairness losses replaced with their losses including $C_*$) shows that $\alpha_{build} \equiv 1$, $\alpha_{fair} \equiv \lambda_{fair} C_{fair}$ and $\alpha_{priv} \equiv \lambda_{priv} C_{priv}$.

To choose $\alpha_i$'s (and by extension $C_*$), (Boyd & Vandenberghe, Section 4.7.5) recommends adjusting the relative "weights" $\alpha_i/\alpha_j$'s. In particular, a point with large curvature of the trade-off function (aka, the knee of the trade-off function) is a good point to reach a compromise between the various objectives; accordingly, finding the $C_*$ that achieves the knee point is recommended.

## D    Broader Impact, Limitations, and Discussion

With the increasing importance of machine learning in sensitive domains, it is crucial to ensure that the machine learning models are trustworthy. However, previous research has primarily focused on addressing a single trust objective at the time or when considering multiple objectives assumed the existence of a central entity responsible for implementing all objectives. We highlight the limitations of this assumption for realistic scenarios with multiple agents and introduce an approach for optimization over multiple agents with multiple objectives (MAMO) to overcome this limitation.

Our approach recognizes the diverse nature of agents involved in deploying and auditing machine learning models. This allows us to make suggestions for guarantee levels that are more likely to

be realizable *in practice*; given that the gains and benefits of different parties have been taken into account. We, however, acknowledge that agents may in fact have a more diverse set of requirements and objectives; and that as a result our models may not be sophisticated-enough to incorporate all such factors. Additionally, we made several assumptions regarding the economic model under which we operate as well as common knowledge of the PF between various objectives. While these assumptions follow established principles in economics (expected utility hypothesis for the former) and in machine learning (the existence of a data-generating distribution for the latter), both are contested in their respective literature.

Finally, we acknowledge that providing "metrics" for human and society values such as fairness and privacy is imperfect at best and fraught with philosophical and ethical issues. Nevertheless, the metrics we used in our study are commonplace in trustworthy ML circles and the search for better, more inclusive, metrics is underway. Our research, therefore, aims to provide systematic guidance on best practices in regulating trustworthy ML practices, and can be adopted for future development in these areas.

From our empirical results, we observe that different ML tasks exhibit different Pareto frontiers (see Section L). As such, an SpecGame played for one task cannot necessarily provide regulation recommendation for other tasks. It remains to be seen how much such recommendations can transfer between tasks even within the same domain (for instance, vision). For instance, recommendation made on the basis of age classification may be ineffective (or too restrictive) for gender estimation.

Finally, we centered our consideration around calculating fines proportional to the privacy and fairness violations of chosen guarantee levels $(\gamma, \varepsilon)$; as well as ensuring they are effective in changing model builder behavior. The converse problem is also important: assuming a bound $C$ on the penalty, what are the maximal $\gamma, \varepsilon$ guarantees that we can expect to be able to enforce?

# F  BACKGROUND ON GAME THEORY

## F.1  BEST RESPONSE

In a multi-agent setup where every agent cares about only one objective (its own), we have a *bi-level optimization* problem. For instance the model builder would be solving:

$$
\begin{aligned}
\min_{\theta_{\text{acc}}} \quad & \ell_{\text{acc}}\left(\theta_{\text{acc}}, \theta_{\text{priv}}, \theta_{\text{fair}}\right) \\
\text{subject to} \quad & \theta_{\text{priv}} = \arg\min_{\theta_{\text{priv}}} \ell_{\text{priv}}\left(\theta_{\text{acc}}, \theta_{\text{priv}}, \theta_{\text{fair}}\right) \\
& \theta_{\text{fair}} = \arg\min_{\theta_{\text{fair}}} \ell_{\text{fair}}\left(\theta_{\text{acc}}, \theta_{\text{priv}}, \theta_{\text{fair}}\right)
\end{aligned}
\tag{8}
$$

Consider $\bar{\theta} = \begin{bmatrix} \theta_{\text{acc}} & \theta_{\text{priv}} & \theta_{\text{acc}} \end{bmatrix}^{\top}$, then we can write the objectives of all agents as

$$
\bar{\ell}(\theta) := \begin{bmatrix} \ell_{\text{acc}}\left(\bar{\theta}\right) & \ell_{\text{priv}}\left(\bar{\theta}\right) & \ell_{\text{fair}}\left(\bar{\theta}\right) \end{bmatrix}^{\top}
$$

. The map $\bar{\ell} : \Theta_{\text{acc}} \times \Theta_{\text{priv}} \times \Theta_{\text{fair}} \mapsto \mathbb{R}^{3}_{\geq 0}$ is vector-valued. Let $\bar{\theta}^*$ be the solution to the bi-level optimization of eq. (8):

$$
\bar{\theta}^* = \arg\min_{\bar{\theta}} \begin{bmatrix} \ell_{\text{acc}}\left(\bar{\theta}\right) & \ell_{\text{priv}}\left(\bar{\theta}\right) & \ell_{\text{fair}}\left(\bar{\theta}\right) \end{bmatrix}
\tag{9}
$$

This formulation allows us to study the interaction of agents whose objectives are defined through a value function (known as the *payoff*). Note that a agent's payoff is not only a function of its own actions, but also those of its peers. This creates an opportunity for the agent to strategize and choose its best possible action given others' actions, where "best" is interpreted as the optimizer of its payoff. These actions form the *best responses* (or BRs) to peers' actions. Therefore, BRs are set-valued mappings from the set of agents' actions onto itself whose fixed points are known as *Nash equilibria*.

More formally, let $BR : \Theta_{\text{acc}} \times \Theta_{\text{priv}} \times \Theta_{\text{fair}} \mapsto \Theta_{\text{acc}} \times \Theta_{\text{priv}} \times \Theta_{\text{fair}}$ be the argmin function of $\bar{\ell}$. This operator calculates the *best response* (BR) of every agent given the choice of parameters $\theta$. $\bar{\theta}^*$ is *fixed-point* of this map: $BR(\bar{\theta}^*) = \bar{\theta}^*$. $\bar{\theta}^*$ is a Nash Equilibrium and eq. (9) describes a game.

### F.2 STACKELBERG COMPETITIONS

Stackelberg competitions model sequential interaction among strategic agents with distinct objectives Fudenberg & Tirole (1991). They involve a leader and a follower. The leader is interested in identifying the best action (BR) assuming rational behavior of the follower. The combination the leader's action and the follower's rational best reaction leads to a strong Stackelberg equilibrium (SSE) Birmpas et al. (2020). This improves over work relying on zero-sum game formulation Yao (1977) where the follower's objective is assumed to be opposed to the leader's objective. An important example for the application of Stackelberg competition in trustworthy ML strategic classification. Therein, strategic individuals can, after observing the model output, adapt their data features to obtain better classification performance. Such changes in the data can cause distribution shifts that degrade the model's performance and trustworthiness on the new data, and thereby requires the model builders adapt their models. In our model governance game framework, the two regulators act as *leaders* while the model builder acts as the *follower*. By following the Stackelberg competition, the model builder aims at obtaining the best-performing ML model given the requirements specified by the regulators.

## G  PARETOPLAY ON FAIRPATE

In FairPATE, we train teacher ensemble models on the training set. These teachers vote to label the unlabeled public data. We then train student models on the now labeled public data. At inference time, the student model does not answer all the queries in the test set. Coverage indicates the percentage of queries that the student does answer.

We denote the student model for classification by $\omega$, the features as $(\mathbf{x}, z) \in \mathcal{X} \times \mathcal{Z}$ where $\mathcal{X}$ is the domain of non-sensitive attributes, $\mathcal{Z}$ is the domain of the sensitive attribute (categorical variable). The categorical class-label is denoted by $y \in [1, \ldots, K]$. We indicates the strategy vector space as $\boldsymbol{s} = (\gamma, \varepsilon)$ where $\gamma$ is the maximum tolerable fairness violation and $\varepsilon$ is the privacy budget.

The loss functions of all agents depend on both $\gamma$ and $\varepsilon$. We normalize all the losses to be between 0 and 1 to ensure loss components (*e.g.*, Equation (3)) are on the same scale. A gradient descent update of $\gamma$ and $\varepsilon$ is:

$$\gamma^t = \gamma^{t-1} - \eta_{\text{fair}} \frac{L}{\partial \gamma}, \ \varepsilon^t = \varepsilon^{t-1} - \eta_{\text{priv}} \frac{\partial L}{\partial \varepsilon} \tag{10}$$

The model builder cares about both student model accuracy and coverage. It would want to provide accurate classification and answer most queries. Its loss function uses a weighted average of the two:

$$\ell_b(\gamma, \varepsilon) = -1 \left( \lambda_b \text{acc}(\gamma, \varepsilon) + (1 - \lambda_b) \text{cov}(\gamma, \varepsilon) \right) \tag{11}$$

where $\lambda_b$ is a hyperparameter set by the model builder that controls how much it values accuracy and coverage. The accuracy and coverage are multiplied with -1 to form the loss because we want to maximize them. Both accuracy and coverage values used are between 0 and 1. At each turn, the model builder decides its response by calculating $\frac{\partial \ell_b}{\partial \gamma}$ and $\frac{\partial \ell_b}{\partial \varepsilon}$ at the current $\varepsilon$ and $\gamma$.

The loss function of the fairness and privacy regulators are $\ell_{\text{fair}}(\gamma, \varepsilon) = \gamma_{\text{ach}}(\gamma, \varepsilon)$ and $\ell_{\text{priv}}(\gamma, \varepsilon) = \varepsilon_{\text{ach}}(\gamma, \varepsilon)$ respectively.

## H  FAIRNESS

We provide more details on the fairness notions used in our empirical study in Section 5.

### H.1  DEMOGRAPHIC PARITY FAIRNESS

Yaghini et al. (2023) adopt the the fairness metric of *multi-class demographic parity* which requires that ML models produce similar success rates (*i.e.*, rate of predicting a desirable outcome, such as getting a loan) for all subpopulations (Calders & Verwer, 2010).

In practice, they estimate multi-class demographic disparity for class $k$ and subgroup $z$ with: $\widehat{\Gamma}(z, k) := \frac{|\{\hat{Y}=k, Z=z\}|}{|\{Z=z\}|} - \frac{|\{\hat{Y}=k, Z \neq z\}|}{|\{Z \neq z\}|}$, where $\hat{Y} = \omega(\mathbf{x}, z)$. They define demographic *parity*

when the worst-case demographic disparity between members and non-members for any subgroup, and for any class is bounded by $\gamma$:

**Definition 4** ($\gamma$-DemParity). *For predictions $Y$ with corresponding sensitive attributes $Z$ to satisfy $\gamma$-bounded demographic parity ($\gamma$-DemParity), it must be that for all $z$ in $\mathcal{Z}$ and for all $k$ in $\mathcal{K}$, the demographic disparity is at most $\gamma$: $\Gamma(z, k) \leq \gamma$.*

## H.2 DISPARATE IMPACT OF DIFFERENTIAL PRIVACY

Esipova et al. (2022) study the disparate impact of privacy on learning across different groups. In particular, they adopt a *accuracy parity* notion of fairness (Bagdasaryan et al., 2019). A fair model, in their view, minimizes the following:

$$\pi(\omega, D_k) = \text{acc}(\omega^*; D_k) - \mathbb{E}_{\tilde{\omega}}[\text{acc}(\tilde{\omega}; D_k)], \tag{12}$$

where $\text{acc}(\omega^*; D_k)$ is accuracy of the optimal accuracy $\omega^*$ on dataset $D_k$ belonging to the $k^{\text{th}}$ subpopulation, and $\tilde{\omega}$ is the privatized model. As with the output of any differentially private mechanism, the accuracy of the privatized model is measured in expectation.

In our experiments, we measure the largest accuracy gap between subgroups in Equation (12). That is, the fairness measure for the regulator in the regulator objective when using DPSGD-Global-Adapt is:

$$\Gamma_{\text{DPSGD-Global-Adapt}}(\omega; D) = \max_k \pi(\omega, D_k). \tag{13}$$

## I ADDITIONAL EXPERIMENTAL SETUP

In all games on FairPATE, we use step sizes $\eta_{\text{fair}} = 10$ and $\eta_{\text{priv}} = 100$, model builder's loss function weightings $\lambda_{\text{fair}} = 0.3$ and $\lambda_{\text{priv}} = 0.3$, step size discount factor $c = 0.67$. In the game on DPSGD-Global-Adapt, we use $\eta_{\text{fair}} = 1$ and $\eta_{\text{priv}} = 5$. All the other game hyperparameters for each game shown in Section 5 are shown in Table 2.

The model architecture and data we use in the experiments follow what is described in the original works for FairPATE Yaghini et al. (2023) and DPSGD-Global-Adapt Esipova et al. (2022). The datasets used for FairPATE and their information are shown in Table 3. For all datasets in FairPATE for the calibration step, we train the student model with Adam optimizer and binary cross entropy loss. We train for 30 epochs with early stopping.

During the games, we put box constraints on the parameters $\boldsymbol{s} = (\gamma, \varepsilon)$ so that they would not be out of range and produce undefined outputs. We use $\gamma \in [0.01, 1]$ and $\varepsilon \in [1, 10]$.

| Figure | Dataset | Algorithm | $\boldsymbol{s} = (\varepsilon, \gamma/\tau)$ | $C_{\textit{fair}}$ | $C_{\textit{priv}}$ |
|---|---|---|---|---|---|
| Figure 2a | UTKFace | FairPATE | (5.0, 0.05) | 0.5 | 0.5 |
| Figure 2b | CelebA | FairPATE | (8.0, 0.01) | 3 | 2 |
| Figure 2c | FairFace | FairPATE | (5.0, 0.05) | 1 | 1 |
| Figure 2a | MNIST | DPSGD-Global-Adapt | (2.895, 0.05) | 1 | 5 |
| Figure 3 | UTKFace | FairPATE | (2.0, 0.01) | 1 | 1 |
| Figure 3 | UTKFace | FairPATE | (9.0, 0.2) | 1 | 1 |
| Figure 4b | UTKFace | FairPATE | (3.0, 0.05) | 1 | 1 |
| Figure 4b | UTKFace | FairPATE | (2.953, 0.067) | 5 | 3 |
| Figure 4b | UTKFace | FairPATE | (7.0, 0.05) | 10 | 10 |
| Figure 4b | UTKFace | FairPATE | (5.980, 0.030) | 3 | 1 |
| Figure 5 | UTKFace, FairFace | FairPATE | (5.0, 0.01) | 3 | 1 |

Table 2: **ParetoPlay hyperparameter settings used in the experiments.**

## J PARETOPLAY WITH INFORMATION ASYMMETRY

A current limitation of our work is the reliance of a common knowledge Pareto frontier (PF). While it has been shown that despite datasets differences, PFs for the same task is similar (and we will

| Dataset | Prediction Task | C | Sens. Attr. | SG | Total | U | Model | Number of Teachers | $T$ | $\sigma_1$ | $\sigma_2$ |
|---------|-----------------|---|-------------|----|-------|---|-------|--------------------|-----|-----------|-----------|
| CelebA | Smiling | 2 | Gender | 2 | 202 599 | 9 000 | Convolutional Network (Table 4) | 150 | 130 | 110 | 10 |
| FairFace | Gender | 2 | Race | 7 | 97 698 | 5 000 | Pretrained ResNet50 | 50 | 30 | 30 | 10 |
| UTKFace | Gender | 2 | Race | 5 | 23 705 | 1 500 | Pretrained ResNet50 | 100 | 50 | 40 | 15 |

Table 3: Datasets used for FairPATE. Abbreviations: **C**: number of classes in the main task; **SG**: number of sensitive groups; **U**: number of unlabeled samples for the student training . Summary of parameters used in training and querying the teacher models for each dataset. The pre-trained models are all pre-trained on ImageNet. We use the most recent versions from PyTorch.

| Layer | Description |
|-------|-------------|
| Conv2D | (3, 64, 3, 1) |
| Max Pooling | (2, 2) |
| ReLUS | |
| Conv2D | (64, 128, 3, 1) |
| Max Pooling | (2, 2) |
| ReLUS | |
| Conv2D | (128, 256, 3, 1) |
| Max Pooling | (2, 2) |
| ReLUS | |
| Conv2D | (256, 512, 3, 1) |
| Max Pooling | (2, 2) |
| ReLUS | |
| Fully Connected 1 | (14 * 14 * 512, 1024) |
| Fully Connected 2 | (1024, 256) |
| Fully Connected 2 | (256, 2) |

Table 4: Convolutional network architecture used in CelebA experiments.

evaluate evaluated ParetoPlay under this setting shortly); it is reasonable expectation that there would always be an information gap between regulators and model builders. While there may be an incentive for builders to close this gap by sharing data (for example, to avoid unwarranted penalties). This is not always possible due to privacy protections for customer data. Furthermore, data sharing assumes that model builders would not misreport despite an incentive to do so for avoiding penalties. We, therefore, argue that instead of trusting the builder, we should seek trust using cryptographic primitives such as multi-party computation (MPC) and zero knowledge proofs (ZKP). A viable future direction then is to assume that instead of model builder and regulator sharing model updates $\omega$ and calculating their Pareto frontiers individually; we allow each party to calculate a Pareto frontier on their own data and share its points $\{(\gamma, \varepsilon, \alpha)\}$ with corresponding proofs for the fairness ($\gamma$) and accuracy loss ($\alpha$), on a zero-knowledge-proven random sample of their dataset. The privacy loss $\varepsilon$ remains a challenge, because an exact DP guarantee requires an audit of the training algorithm. We can however, establish an upper bound on the privacy loss of the model, if we can ensure that a particular sample (known as a privacy canary) has been used in the training of the model. The creation of the privacy canary, and providing å formal proof of it are open research problems.

**Empirical Evaluation.** We show results for a game where different agents have access to different dataset. This is a realistic assumption because private data would not normally be shared between agents. In this setup, the regulators have access to FairFace, whereas the model builder has access to UTKFace. The agents use their respective dataset to form their loss functions. During calibration, they train models on their own datasets as well. The results of the game is shown in Figure 5. We observe that although the agents use different datasets, the game is able to converge and both the privacy and fairness constraints are satisfied. This shows that it is possible to reach desirable equilibria even without all agents having access to the same dataset.

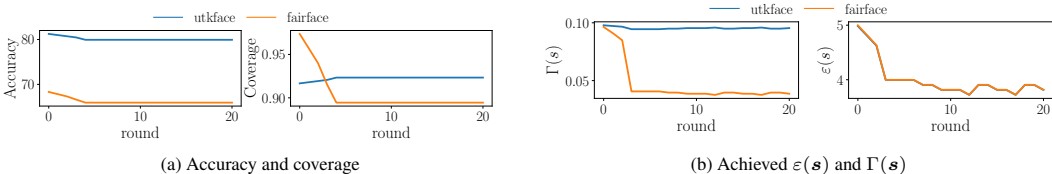

(a) Accuracy and coverage            (b) Achieved $\varepsilon(\boldsymbol{s})$ and $\Gamma(\boldsymbol{s})$

Figure 5: **Agents need not have access to the same dataset to achieve desired equilibria** We simulate a game where regulators have access to FairFace and model builder has access to UTKFace. We observe the game still converges and the trust constraints are both satisfied.

## K    PROOFS

### K.1    CONVERGENCE OF PARETOPLAY

We provide a proof sketch for the converge of ParetoPlay. We consider Algorithm 1 and its iterations up to convergence to an equilibrium $\boldsymbol{s}^*$; that is, we assume penalty scalers are not-adjusted mid-play for incentive design purposes (See discussion in Section 5). This does not reduce the generality of our claims here, since every adjustment of penalty scalers would force a new run of the algorithm which as we will show is convergent.

Furthermore, we instantiate the SpecGame which ParetoPlay simulates using the FairPATE learning framework of [44]. Apart from the particularities of each learning framework reflected in their loss terms, the analysis presented here should apply to other learning frameworks (such as DP-SGD methods).

We assume that the market is regulator-led, meaning that the regulators have already chosen guarantee levels $\varepsilon_0$ and $\gamma_0$. Since $C_*$ and $\lambda_*$ are both scalers, W.L.O.G., we assume $\lambda_{\text{fair}} = \lambda_{\text{priv}} = 1$, then

$$F := L_{\text{build}}(\gamma, \varepsilon) = \ell_{\text{build}}(\gamma, \varepsilon) + C_{\text{fair}}\ell_{\text{fair}}(\gamma) + C_{\text{priv}}\ell_{\text{priv}}(\varepsilon) \tag{14}$$

For a builder using PATE-based methods, $\ell_{\text{build}}$ is the PATE-student average loss which is a typical neural network training loss (such as cross-entropy).

The Pareto frontier is a trade-off function between acc-priv-fairness; so it is enough to be looking at $\ell_{\text{build}}(\gamma, \varepsilon)$ exclusively. We can show that Algorithm 1 is similar to minimization with subgradient methods with square summable but not summable step sequences (Boyd et al.). If we can claim (*) is $L$-smooth, then we can use the standard argument in Boyd et al., Section 2 to show convergence.

**Theorem 1.** *ParetoPlay recovers a Correlated Nash Equilibrium of the SpecGame.*

*Proof.* Assuming all agents play on the Pareto frontier (Algorithm 1), we need to show that there is no incentive to deviate from playing on the Pareto frontier.

Assume that Builder (B) reports a $\omega_r$ that is not on the PF. Assume there that there exists some $\omega^*$ on the PF, this means that $\omega^*$ Pareto dominates $\omega_r$: it is at least as good in all objectives, and better at least in one. We first note that reporting $\omega_r$ where $\ell_{acc}(\omega_r) > \ell_{acc}(\omega^*)$ is irrational (in the game theoretic sense that it increases the agent's cost instead of reducing it) and thus never a best response for B. So we can only cases where it holds that either $\ell_{acc}(\omega_r) < \ell_{acc}(\omega^*)$ and $\ell_{fair}(\omega_r) > \ell_{fair}(\omega^*)$, or $\ell_{acc}(\omega_r) < \ell_{acc}(\omega^*)$ and $\ell_{priv}(\omega_r) > \ell_{priv}(\omega^*)$, or both hold. But every agent in Pareto Play, re-calculates its Pareto frontier as a first step (Line 2 in Alg. 1). Assume, if at time $t-1$, B adds $\omega_r$ to $R^{(t)}$. At time $t$, the regulator would re-calculate its PF; but since $\omega_r$ is not on the PF, either a) some other $\omega_*$ already exists in $R^{(t)}$ which dominates $\omega_r$, and therefore $\omega_r$ never appears in the rest of the regulators round; or b) if no such $\omega_*$ exists, the regulator will use $\omega_r$ as initialization, do not change the penalty scale in Line 6 (again because it would be irrational for B to report a $\omega_r$ which would cause a penalty), and take a step on the Pareto frontier to improve the corresponding regulator loss. At this point, depending on which objective value was under-reported by B, the regulator would either be able to find an $\omega_*$ that Pareto dominates $\omega_r$ —at which point $\omega_r$ again is effectively removed from the PF calculations—or the next regulator is going to make a

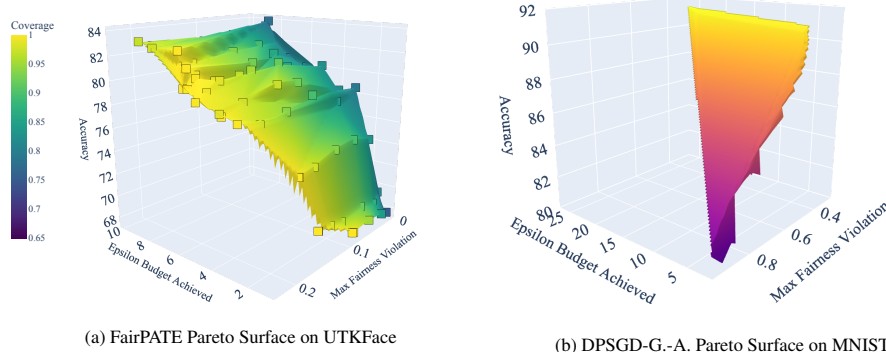

(a) FairPATE Pareto Surface on UTKFace

(b) DPSGD-G.-A. Pareto Surface on MNIST

Figure 6: Pre-computed Pareto Frontier surfaces.

gradient step and find the appropriate $\omega_*$ that Pareto dominates the misreported $\omega_r$. In the worst-case where we lose gradient information (in a boundary condition, or near an inflection point), we note that every agent trains a model in the Calibration phase (line 10). At this point, with a near 0 gradient step, $\omega_* \approx \omega_r$ is reevaluated by one of the regulators, which ensures that $\ell_{priv}$ and/or $\ell_{fair}$ values are corrected, which again leads to exclusion of $\omega_r$ from the Pareto frontier. Therefore, we have shown that there is no incentive to play a Pareto inefficient solution. □

## L  PARETO FRONTIERS

In Figure 6, we highlight the Pareto frontiers over which ParetoPlay is played that are experimental results in Section 5 demonstrate.

