# OpenReview forum: "Regulation Games for Trustworthy Machine Learning"
_ICLR.cc/2024/Conference — Submitted to ICLR 2024_

### Official Review · Reviewer_wL1U · 2023-10-27

**Soundness:** 2 fair
**Presentation:** 2 fair
**Contribution:** 2 fair
**Rating:** 6
**Confidence:** 3

**Summary:**

The paper studies the trade-off in trustworth ML, specifically that between fairness, privacy and model utility, by formulating it as a multi-agent game (called SpecGame) among two regulators and a model builder. The authors design a method called ParetoPlay to search for an equilibrium on the Pareto frontier. Experiments show that the designed method can be instantiated to two existing trustworth ML algorithms and demonstrate the trade-off between fairness vs. privacy vs. utility achievable by the designed method.

**Strengths:**

- The problem of trade-off among different important criteria in trustworth ML is important.

- The formulation of the interaction as a multi-agent game is intuitive.

- The theoretical analysis and empirical results demonstrate the interaction among the agents in the game, which can be useful in understanding the (possible fundamental) limitations when designing new trustworth ML algorithms.

**Weaknesses:**

- A key assumption is the common-knowledge of a pre-calculated PF among the
considered critera, specifically privacy, fairness and model utility. This can be difficult to satisfy in practice.

- There are several (simplifying) assumptions made (which can take away the pratical feasibility of the work). For two examples,
    - > We assume that regulators are able to give penalties for violations of their respective objective which they formulate as a utility (or value) function.

    - > We assume the regulators hold necessary information about the task at hand in the form of a Pareto Frontier (PF) which they use to choose fairness and privacy requirements that taken together with the resulting accuracy loss are Pareto efficient:

- The writing can be improved (for details, see the questions below).

**Questions:**

1. In Section 1
    > This is because nowadays ML models are trained, maintained, and audited by separate entities—each of which may pursue their own objectives.

    Are there references or real-world use-cases where this is true or implemented?

2. In Section 1,
    > ... that assumes shared knowledge of a pre-calculated PF between agent objectives.

    Can this PF be realized in practice? i.e., how to accurately obtain it? and if only a somewhat inaccurate one can be obtained, what are its implications?


3. What is $i\in I$ in Definition 1?

4. In Section 3,

    > In this work, we do not consider a competition between regulatory bodies since both are assumed to be governmental agencies.

    Even though the regulatory bodies are not set out to compete with each other, the inherent tension between the objectives can lead to competitive strategy profiles and actions, right?

    In that case, what is the significant distinction of "not considering a competition between the regulatory bodies"?


5. What is $\\{c_{i}^{(t)} \\}^t$ in the overall discounted loss in Section 3.2 ?

6. In Section 4,
    > However, ...the highly non-convex nature of agent losses in $\texttt{SpecGame}$

    How is the non-convexity addressed?

7. In Section 4,

    > for $t>1$, we assume a mapping from the penalty values in S to trustworthy parameters values $s_{reg} = (\gamma, \epsilon)$

    (How) can this assumption be satisfied in practice ?

8. What do the colors represent in Figure 4a?

9. Are the experimental results averaged over multiple trials? If so, is there an analysis of the variation?

**Details Of Ethics Concerns:**

N.A.

---

> ### Author Response · Authors · 2023-11-17
> **Response to Weaknesses**
>
> We thank the reviewer for their feedback. Below we address each point raised in "Weaknesses" section.
>
> - A key assumption is the common-knowledge of a pre-calculated PF among the considered critera, specifically privacy, fairness and model utility. This can be difficult to satisfy in practice.
>
> 	- Please see the general response.
>
> - There are several (simplifying) assumptions made (which can take away the pratical feasibility of the work). For two examples,
>   > We assume that regulators are able to give penalties for violations of their respective objective which they formulate as a utility (or value) function.
>
> 	- Despite nascent privacy laws, there are already [many instances](https://www.enforcementtracker.com) of monetary penalties levied for violation of data privacy laws such as GDPR. Therefore, modeling such fines to be proportional to the privacy loss (as we do) is not a limitation beyond the fact that we take differential privacy as our notion of privacy—a choice that we believe is presently justified because differential privacy is the dominant privacy notion for machine learning models.
>
> - > We assume the regulators hold necessary information about the task at hand in the form of a Pareto Frontier (PF) which they use to choose fairness and privacy requirements that taken together with the resulting accuracy loss are Pareto efficient:
>
> 	- Please see the general response.

---

> ### Author Response · Authors · 2023-11-17
> **Response to Questions (Part 1)**
>
> - In Section 1
> > This is because nowadays ML models are trained, maintained, and audited by separate entities—each of which may pursue their own objectives.
> Are there references or real-world use-cases where this is true or implemented?
> 	- In California, the Consumer Privacy Act in California stipulates that the Attorney General enforces the privacy of employment background screening.  However, the Employment Non-Discrimination Act is enforced by the Department of Justice. These are two separate governmental entities. We believe cross-agency cases such as this should be more prevalent as the use of automated decision making systems grows. Having said that, we are not trained in law so we refrain from making a case reference.
>
> - In Section 1,
>  > ... that assumes shared knowledge of a pre-calculated PF between agent objectives.
> Can this PF be realized in practice? i.e., how to accurately obtain it? and if only a somewhat inaccurate one can be obtained, what are its implications?
> 	- Yes, it is possible to develop a cryptographic zero-knowledge-proof (ZKP) version of ParetoPlay where the agents not only share models but also proof of trustworthiness guarantees  (see Shamsabadi et al. 2022 for an example of a ZKP proof of fairness). We expand on this idea in Section J in the appendix, however, this is beyond the scope of the present work which lays the game theoretic foundations for ML Regulation.
>
> 	- For a discussion on the shared Pareto frontier, please see the general response.
>
> - What is  $i \in I$ in Definition 1?
>
> 	- $I$ is the set of objectives.  This was left out by mistake and has been added to Definition 1.
>
> - In Section 3,
> > In this work, we do not consider a competition between regulatory bodies since both are assumed to be governmental agencies.
> Even though the regulatory bodies are not set out to compete with each other, the inherent tension between the objectives can lead to competitive strategy profiles and actions, right?
>
> 	- Yes, that is correct. But the regulators being *governmental* agencies, the higher level agent (the government) can make a decision regarding the fairness and privacy trade-off. We capture this scenario in the first stage of a regulator-led SpecGame in the 2nd paragraph of Section 4.1.
>
> - In that case, what is the significant distinction of "not considering a competition between the regulatory bodies"
>
> 	- An alternative game formulation, for example, can consider the regulatory bodies as independent agents (not controlled by a 3rd agent, such as the government).
>
> 	- In this setting, the regulators can fight against each other to improve their own metric, possibly to the detriment of the general good. For instance, low-privacy low-fairness can be an equilibrium of such a game, which does not benefit either party. In similar 2-player games, a solution to this is to use a third party mediator to ensure a better equilibrium (see Section 10.7.3 in Shoham and Leyton-Brown 2009 for instance).
>
> 	- Note that in our formulation, we already have a natural candidate for such a mediator which is the the government. We have thus shown that a solution to the joint game and subgame reduces to our original game formulation.
>
> - What is $B: =\left\\{c_i^{(t)}\right\\}^t$   in the overall discounted loss in Section 3.2 ?
>
> 	- This is a factor that depends on how the agent perceives its current loss w.r.t its past losses. In the most general case can change over time, $B = B(t)$ ; which is what the expression above models. We have reformulated this general loss as $L_i=\\sum_{t=1}^{\\infty}\\left\\{\\prod^{t}_{i=1}c_i^{(t)}\\right\\} L_i^{(t)}$ which should make this point clearer. Note that, as before, we assume $B$ is constant with time  $c_i^{(t)} = c$ which is a common assumption in modeling of repeated games.

---

> ### Author Response · Authors · 2023-11-17
> **Response to Questions (Part 2)**
>
> - In Section 4,
> > However, ...the highly non-convex nature of agent losses in
> How is the non-convexity addressed?
>
> 	- By approximating and interpolating the agent losses on Pareto frontier (as we discuss in paragraph titled **Approximation and calibration** in Section 4.1).
>
> - In Section 4,
> > for $t>1$ , we assume a mapping from the penalty values in $\mathrm{S}$ to trustworthy parameters values $s_{, u y}=(\gamma, \epsilon)$
> (How) can this assumption be satisfied in practice ?
>
> 	- What this assumption says is that for a given pair of parameters $\boldsymbol{s}=(\gamma, \epsilon)$ we can find a penalty to enforce it. Note that this is trivially true for a large enough penalty $C_\text{fair}$ ( $C_\text{priv}$ ) ensuring that fairness (privacy) losses dominate the model builders loss, and loss update in equations 3 and 4. Therefore, for any feasible solution to the joint optimization of training loss under fairness and privacy constraints, this assumption holds; and therefore does not limit the applicability of our algorithm in practice.
>
> - What do the colors represent in Figure 4a?
>
> 	- The colors in Figure 4a represent different $C_{fair}$ values. The lightest shade correspond to the lowest $C_{fair}$ , and the darkest the highest. We have adjusted the figure legend and its caption to make this more clear.
>
> - Are the experimental results averaged over multiple trials? If so, is there an analysis of the variation?
>
> 	- We run the games on pre-computed Pareto frontiers which we have calculated over many runs of the private and fair learning models we have adopted (FairPATE and DPSGD-Global-Adapt) to ensure we have an accurate Pareto frontier. As a result,  the only source of randomization in running ParetoPlay would be the calibration step, where the agents train a model to re-compute the Pareto surface. This pre-calculation of trustworthiness hyper-parameters makes for a relatively smooth (but non-convex) Pareto surface (see Figure 6 in the appendix). As a result we, we have found out that calibration often does not produce a new point on the Pareto surface  and therefore we have not observed varied convergence results. The different convergence results we see in the experiments are due to having different starting points and different regulator penalty scalar $C_{fair}$ and $C_{priv}$ .
>
>
> We once again thank the reviewer for their detailed feedback and kindly ask them to consider raising their score if we have addressed their concerns successfully. We are happy to answer further questions.

---

> > ### Comment · Reviewer_wL1U · 2023-11-21
> > **Post rebuttal**
> >
> > I thank the authors for their detailed response, especially the the supporting evidence and examples.
> >
> > Most of my questions are addressed. While there are limitations (unavoidable) such as those pointed out in the general comment and differential privacy as the main choice of privacy, I believe that this paper meets the bar of acceptance and hence raise my score to 6.

---

> > > ### Author Response · Authors · 2023-11-21
> > > **Thank you for engaging with us!**
> > >
> > > We thank the reviewer for their consideration and for raising their score. We are ready to answer any further questions that the reviewer may have.

---

### Official Review · Reviewer_kbV4 · 2023-11-01

**Soundness:** 1 poor
**Presentation:** 2 fair
**Contribution:** 1 poor
**Rating:** 3
**Confidence:** 3

**Summary:**

The paper is concerned with trustworthy ML. The main contribution is to model the setting as a game (SpecGame) between the model builder and the regulator who is interested in fairness and privacy. An algorithm ParetorPlay is introduced and it is shown that the agents remain on the Pareto frontier.

**Strengths:**

-The idea seems interesting and novel and one can think of it as modeling a wide set of problems.

-The paper is also concerned with an important problem (trustworthy ML).

**Weaknesses:**

1-I think the main contribution of the paper is to model the dynamic interaction between the model builder and the regulator. Accordingly, it is more reasonable to think of only fairness or privacy or to possibly even abstract/lump both issues into one. I don't see how having these two considerations has added to the model. One can also consider the safety of the model or its robustness to adversarial manipulations as part of the regulator's concern for example.

2-Why is the paper searching for Pareto Optimality instead of a Nash Equilibrium? Both agents (builder or regulator) are interested in their utility and as a result would deviate to increase it which is what would be captured by a Nash Equilibrium

3-The section “Making a uniform strategy space.” on page 6 is very unclear. What does "consistent" mean? It seems to suggest that the strategies are fixed. Further, it seems that the horizon is n. If that is the case the why does the utility in section 3.2 sum to infinity?

4-Why is a correlated equilibrium and correlation device well-motivated in this setting? Also the paper mentions that “ This is known as a correlation device. If playing according to the signal is a best response for every player, we can recover a correlated equilibrium (CE). We leave the theoretical proof of this conjecture to future work.” But doesn’t Theorem 1 prove that you have a correlated equilibrium so is it a conjecture?

**Questions:**

Please see Weaknesses above.

---

> ### Author Response · Authors · 2023-11-17
> **Response**
>
> We thank the reviewer for their feedback. Below we address each point raised:
>
> - 1-I think the main contribution of the paper is to model the dynamic interaction between the model builder and the regulator. Accordingly, it is more reasonable to think of only fairness or privacy or to possibly even abstract/lump both issues into one. I don't see how having these two considerations has added to the model. One can also consider the safety of the model or its robustness to adversarial manipulations as part of the regulator's concern for example.
>
> 	- We agree with the reviewer that our framework is general and can include other trustworthy objectives in future work that builds on our framework.
>
> 	- The privacy and fairness regulators can be lumped into one. In fact, we do so at the start of a regulator-led game (in Section 4.1). There we note that considering a joint regulator  not resolve the inherent conflicts between the two objectives (fairness and privacy).
> 	  **In practice ML models may be audited by separate entities. Therefore, our goal in keeping the regulators separate was to consider broader and more realistic scenarios.** For instance, the Consumer Privacy Act in California stipulates that the Attorney General enforces the privacy of employment background screening. However, the Employment Non-Discrimination Act is enforced by the Department of Justice. A joint regulator tacitly assumes that authorities would coordinate in every decision which is an impractical assumption.
> - 2-Why is the paper searching for Pareto Optimality instead of a Nash Equilibrium? Both agents (builder or regulator) are interested in their utility and as a result would deviate to increase it which is what would be captured by a Nash Equilibrium
>
> 	- The Nash Equilibria (NE) are not necessarily optimal outcomes for the society. For example, in the US there are fairness regulations that penalize employers for discriminative hiring behavior whereas privacy laws are limited to a few states. This discrepancy can lead the builders to create models that are fair but non-private. The corresponding SpecGame is in a Nash equilibrium $\boldsymbol{s}_1$ because no single party has any alternative strategy that can improve its outcome.
>
> 	- Now consider an alternative strategy profile $\boldsymbol{s}_2$ where both fairness and privacy regulators could see an improvement while not hurting the builders utility. Between the two, $\boldsymbol{s}_2$ is a societally optimal strategy profile; and indeed, $\boldsymbol{s}_2$ Pareto dominates $\boldsymbol{s}_1$ .
>
> 	- Given that ML regulation intends to achieve societal optimality we search for Pareto efficient points.
>
> - 3-The section “Making a uniform strategy space.” on page 6 is very unclear. What does "consistent" mean? It seems to suggest that the strategies are fixed. Further, it seems that the horizon is n. If that is the case the why does the utility in section 3.2 sum to infinity?
>
> 	- By consistent, we mean that the domain of strategies for each stage of the game is the same. Furthermore, the SpecGame is in the class of infinitely-repeated Stackelberg games; therefore $n \rightarrow \infty$ .
>
> 	- We have added clarification for both comment to the main text.
>
> - 4-Why is a correlated equilibrium and correlation device well-motivated in this setting? Also the paper mentions that “ This is known as a correlation device. If playing according to the signal is a best response for every player, we can recover a correlated equilibrium (CE). We leave the theoretical proof of this conjecture to future work.” But doesn’t Theorem 1 prove that you have a correlated equilibrium so is it a conjecture?
>
> 	- We apologize for the confusion. Indeed the sentence "We leave the theoretical proof of this conjecture to future work." was mistakenly left in the section from a prior submission where we did not have a formal proof).
>
>
> We once again thank the reviewer for their detailed feedback and kindly ask them to consider raising their score if we have addressed their concerns successfully. We are happy to answer further questions.

---

> > ### Comment · Reviewer_kbV4 · 2023-11-22
> >
> > I thank the authors for the feedback. I will keep my score as it is.

---

### Official Review · Reviewer_SMAr · 2023-11-06

**Soundness:** 3 good
**Presentation:** 2 fair
**Contribution:** 3 good
**Rating:** 5
**Confidence:** 4

**Summary:**

This paper studied the problem of multi-agent and multi-objective machine learning (ML) regulation games where there exist separate regulators enforcing privacy and fairness constraints on the learning model. Prior work in trustworthy ML often implicitly assumes that a single entity is in charge of implementing these different objectives, which is not realized in practice. The authors instead proposed SpecGame, a general framework for ML regulation games between three agents: model builder, fairness regulator, and privacy regulator. Since the agents' privacy loss is difficult to estimate in post-processing, the authors also proposed ParetoPlay, i.e. using a pre-calculated Pareto frontier as common knowledge among all agents, to simulate the interactions between agents in a SpecGame and recover equilibria points. Finally, the authors provided experimental results to show the suboptimality of the single-agent model in trustworthy ML and answer questions on how the regulators can achieve the desired equilibrium by changing their incentives after the game has converged.

**Strengths:**

Strengths:
- The proposed model of multi-agent multi-objective is novel and interesting to study.

- The proposed scenario of SpecGame is well-defined and makes sense.

**Weaknesses:**

Weaknesses:
- The assumption of a pre-calculated Pareto frontier as common knowledge does not have theoretical proofs and the discussion around the empirical evaluation in Appendix J is lacking.

- Throughout the main body, the authors sometimes refer to notations that were not defined previously. For example, in Section 3.2, notation c_i^{(t)} and L_i^{(t)} are the first time the "(t)" superscript is used. In Equation 4, the notation \nabla_s is used without a definition.

- The discussion of the experiments in Section 5 is confusing. At the end of Section 1, the author claimed that the experiments would highlight the suboptimality of studying trustworthy ML in a single-agent framework. However, in Section 5, the first research question shows that multi-agent setup leads to sub-optimalities.

- Figure 1 does not have a legend. Overall, most figures are hard to read and the captions do not provide sufficient description of the experiments.

- Minor typos: \ell_build(w) instead of \ell_b(w) on page 5 under Equation 3.

**Questions:**

- Can the authors clarify the assumption of a pre-calculated Pareto frontier as common knowledge?
- Can the authors clarify the discrepancy between RQ1 and the contribution claimed at the end of Section 1?
- Can the authors provide a more detailed description of the experiments in Section 5, as well as the reasoning for choosing the hyperparameters described in Appendix I? Particularly, the step size discount factor and the loss function weightings \lambda_fair and \lambda_priv?

---

> ### Author Response · Authors · 2023-11-17
> **Response to Weaknesses**
>
> We thank the reviewer for their feedback. Below we address each point raised in "Weaknesses" section.
>
> - The assumption of a pre-calculated Pareto frontier as common knowledge does not have theoretical proofs and the discussion around the empirical evaluation in Appendix J is lacking.
>
> 	- Please see the general response.
>
> - Throughout the main body, the authors sometimes refer to notations that were not defined previously. For example, in Section 3.2, notation $c_i^{(t)}$ and $L_i^{(t)}$ are the first time the "(t)" superscript is used. In Equation 4, the notation $\nabla_s$ is used without a definition.
>
> 	- We have added the clarification of all the aforementioned cases (marked in blue in Section 3.2 and Eq. 4)
>
> - The discussion of the experiments in Section 5 is confusing. At the end of Section 1, the author claimed that the experiments would highlight the suboptimality of studying trustworthy ML in a single-agent framework. However, in Section 5, the first research question shows that multi-agent setup leads to sub-optimalities.
>
> 	- We agree with the reviewer that wording of RQ1 was confusing. We have re-written it for clarity to:
> 	  > (RQ1) if regulators produced specifications for trustworthy parameters $\boldsymbol{s}$ using a single-agent-optimized model, would that recommendation be respected in the multi-agent setting?
>
> 	  and the description of the results has also been updated to:
> 	  > Single-agent-optimized specifications lead to sub-optimalities in the multi-agent settings
> - Figure 1 does not have a legend. Overall, most figures are hard to read and the captions do not provide sufficient description of the experiments.
>
> 	- We have addressed the reviewer's comment and re-wrote the captions for all experiment figures for clarity.
>
> - Minor typos: ell_build(w) instead of ell_b(w) on page 5 under Equation 3.
>
> 	- We thank the reviewer for their attention. We have fixed the typo.

---

> ### Author Response · Authors · 2023-11-17
> **Response to Questions**
>
> - Can the authors clarify the assumption of a pre-calculated Pareto frontier as common knowledge?
>
> 	- Please see the general response.
>
> - Can the authors clarify the discrepancy between RQ1 and the contribution claimed at the end of Section 1?
>
> 	- We answered this question in the previous section.
>
> - Can the authors provide a more detailed description of the experiments in Section 5, as well as the reasoning for choosing the hyperparameters described in Appendix I? Particularly, the step size discount factor and the loss function weightings λ_fair and λ_priv?
>
> 	- We have added a more detailed description of the results as well as more informative captions to the paper (as per reviewer's prior comments). We would be happy to address any other specific confusions about the results.
>
> 	- **Regarding the discount factor.** The $c$ discount factor is one of three scalers which control the convergence behavior; with the other two being $C_*$ and $\lambda_*$ . The discount factor $c < 1$ ensures convergence of the repeated game (as we discuss in Section 3.2). With other factors held constant, as $c$ decreases the dynamics remain the same, but the game will be extended in time.
>
> 	- **Regarding penalty scalers.** $\lambda_*$ are scalars that make the (monetary) penalties comparable to the model loss. Note that $\lambda_*$ s are private information to builder (e.g,  1 million privacy fine compares to the gains from producing a 1-loss-unit lower loss from the trained model). Since regulators do not observe $\lambda_*$ they adjust the impact of their penalties using $C_*$ . In summary,  $\lambda_*$ and  $C_*$ both control the game dynamics but since we seek policy guidance for regulators and those only control $C_*$ we set $\lambda_*=1$ and run ablations with $C_*$ .
>
>
> We once again thank the reviewer for their detailed feedback and kindly ask them to consider raising their score if we have addressed their concerns successfully. We are happy to answer further questions.

---

> ### Author Response · Authors · 2023-11-22
> **As the author interaction is coming to an end, we are waiting for your post-rebuttal response.**
>
> Dear reviewer,
>
> Thank you for your feedback on our work. Your suggestions have improved our work. As the author interaction is coming to an end, we are waiting for your post-rebuttal response. If our response answers your concerns, please consider raising your scores. If not, please let us know of your further questions such that we can answer them in the remaining time.
>
> We thank you for all your efforts!

---

### Author Response · Authors · 2023-11-17
**General Author Response**

We thank the reviewers wholeheartedly for their detailed comments and constructive feedback.

- We are glad that all the reviewers found our problem (trustworthy ML) important and our game theoretic formulation of it novel and interesting.
- In particular, reviewers found the framework intuitive, general and amenable to modeling a wide set of problems in trustworthy ML; and our proposed instantiation of it, SpecGame, to be well-defined.
- One reviewer highlighted the importance of discovering the trade-offs among different trustworthy criteria and  found our theoretical and empirical analysis of agent interaction adopting these criteria to be useful for understanding the possible fundamental limitations of new trustworthy ML algorithms.

**New Manuscript and Color Codings.** We have applied reviewer's comments within the manuscript and have highlighted newly added parts in "blue" and fixed typos in "red". In order to fit the new manuscript within 9 pages we have also shortened the last contribution point which is marked in "teal".

We will address reviewers shared concern regarding the assumption of a shared Pareto frontier, and new policy developments since submission that we believe lends more credence to it.

**Assumption of Shared Pareto Frontier and Information Symmetry**

The reviewers expressed concerns regarding  the assumption of a shared Pareto frontier. We think, however, that the root cause of these concerns is the **information asymmetry** that inevitably exists between regulators and model builders regardless of the modeling assumptions.

We addressed these concerns from a technical standpoint in Section J with **an experiment where regulators and model builders have access to completely different datasets under the same task** and showed that ParetoPlay converges and can still recover useful equilibria. Furthermore, we provided guidance for future work on adding cryptographic zero-knowledge-proof (ZKP) of fairness and privacy interventions to ParetoPlay which would circumvent the information asymmetry issue.

From a policy point of view, since the submission of our work, there has been important developments in the ML Regulation space that beckons a future where the assumption of information symmetry is more realistic.

**Recent policy development makes the Information Symmetry more pertinent**

On October 30, 2023, the US administration has issued an [Executive Order](https://www.whitehouse.gov/briefing-room/presidential-actions/2023/10/30/executive-order-on-the-safe-secure-and-trustworthy-development-and-use-of-artificial-intelligence/) which has called on "independent regulatory agencies" (i.e., our regulators) to  "emphasize and clarify" responsibilities of the "regulated entities" (i.e. our model builders) regarding the fairness and privacy of their models:

> **Section 8.b.(i):**
(C)  incorporation of equity principles in AI-enabled technologies used in the health and human services sector, using disaggregated data on affected populations and **representative population data sets when developing new models, monitoring algorithmic performance against discrimination and bias in existing models**, and helping to identify and mitigate discrimination and bias in current systems;
(D)  **incorporation of safety, privacy, and security standards into the software-development lifecycle** for protection of personally identifiable information, including measures to address AI-enhanced cybersecurity threats in the health and human services sector;

Since then, the issue of safety and ML trustworthiness has become ever more prominent in the public sphere. For instance, [President Obama in a recent interview said](https://youtu.be/X15o2sG8HF4?t=564)
> I think it's entirely appropriate then for us to plant a flag and say, "...frontier companies, you need to disclose what your safety protocols are... Tell us what tests you're using. Make sure that we have some independent verification that right now this stuff is working."

In summary, although these are early days for ML Regulation, from a policy point of view our assumption of information symmetry, whether enforced through transparency laws or audits by a [new agency](https://www.nytimes.com/2023/07/27/opinion/lindsey-graham-elizabeth-warren-big-tech-regulation.html) seems increasingly closer to reality and we believe this makes our submission even more timely than before.

We are happy to further engage with reviewers as part of the discussion phase and hope that the reviewers consider raising their scores.

---

> ### Author Response · Authors · 2023-11-21
> **Thank you for your feedback. Please consider our rebuttal responses.**
>
> Dear reviewers,
>
> Thank you for your valuable feedback. We believe it has improved our submission.  Given that the author interaction window is coming to an end, we would like to kindly ask you to consider the rebuttal responses and let us know if they address your concerns and, if they do, to consider raising your scores.
>
> It would be our pleasure to answer any further questions.

---

### Meta-Review · Area_Chair_eBqQ · 2023-12-11

**Metareview:**

This paper studies multi-agent multi-objective modeling in the context of trustworthy machine learning (ML). All reviewers agree that this paper is aligned with an interesting direction of trustworthy ML. However,  a key concern raised by all reviewers is the assumption of a pre-calculated Pareto Frontier (PF) and its practical feasibility. We believe this was not fully addressed in the response. The reviewers consistently mention issues with clarity and consistency in the paper.  The reviewers also raised questions about the experimental setup. The authors should consider providing a more detailed description of the experimental setup, including the rationale behind the choice of hyperparameters in subsequent revisions.

**Justification For Why Not Higher Score:**

The paper does not meet the bar for acceptance.

**Justification For Why Not Lower Score:**

N/A

---

### Decision · Program_Chairs · 2024-01-16

Reject